Evaluation of colonization and mutualistic endophytic symbiosis of Escherichia coli with tomato and Bermuda grass seedlings

Verma Satish K. 1 skvermabhu@gmail.com
http://orcid.org/0000-0002-8672-2279 Chen Qiang 2
White James Francis 2 jwhite3728@gmail.com
1 Department of Botany, Institute of Science, Banaras Hindu University , Varanasi, Uttar Pradesh , India
2 Department of Plant Biology, Rutgers University , New Brunswick, New Jersey , United States of America
Gillespie Joseph
Electronic publication date: 2022 Aug 10
Publication date: 2022
Volume: 10
Electronic Location ID: e13879
Received 2022 Mar 24; Accepted 2022 Jul 19
Copyright: © 2022 Verma et al.
Copyright year: 2022
Copyright holder: Verma et al.
License: This is an open access article distributed under the terms of the Creative Commons Attribution License, which permits unrestricted use, distribution, reproduction and adaptation in any medium and for any purpose provided that it is properly attributed. For attribution, the original author(s), title, publication source (PeerJ) and either DOI or URL of the article must be cited.
License URL: https://creativecommons.org/licenses/by/4.0/

Keywords: Plant growth promotion, Escherichia coli, Seedlings, GFP, Fluorescence microscopy

Funding: UGC, India, IoE BHU USDA-NIFA Multistate Project W4147 New Jersey Agricultural Experiment Station This work is supported by UGC, India, IoE BHU (Incentive grant), and also supported by USDA-NIFA Multistate Project W4147, and the New Jersey Agricultural Experiment Station. The funders had no role in study design, data collection and analysis, decision to publish, or preparation of the manuscript.

==============================
Escherichia coli is generally considered a bacterium associated with animal microbiomes. However, we present evidence that E. coli may also mutualistically colonize roots of plant species, even to the extent that it may become endophytic in plants. In this study we used GFP tagged E. coli to observe its colonization and effects on tomato (Solanum lycopersicum) and Bermuda grass (Cynodon dactylon) seedling development and growth. Inoculation with the bacterium significantly improved root development of both seedlings tested. Treatment also increased the photosynthetic pigments in Bermuda grass seedlings. However, effects on shoot length in both seedlings were not significant. This bacterium was found to produce indole acetic acid (IAA) up to 8.68 ± 0.43 µg ml−1 in the broth medium amended with tryptophan. Effects on seedling root growth could, in part, be explained by IAA production. Bacteria successfully colonized the root surfaces and interiors of both seedlings. Tagged bacteria expressing the GFP were observed in the vascular tissues of Bermuda grass seedling roots. Seedlings with bacteria showed greater survival and were healthier than seedlings without bacteria, indicating that E. coli set up a successful mutualistic symbiosis with seedlings. E. coli is not commonly considered to be a plant endophyte but is more generally considered to be a crop contaminant. In this study we show that E. coli may also be an endophyte in plant tissues.

Introduction

Escherichia coli is generally considered to be a bacterium associated with animal microbiomes, in particular intestinal microbiota. E. coli is typically not a soil bacterium as it generally reported in faecal matter pollution in natural environments including soil and water (Blount, 2015). Although E. coli is not a natural inhabitant of the soil or rhizosphere, it has been reported to grow in different soil types ranging from tropical to temperate regions (Oliver et al., 2006; Semenov et al., 2008; Nautiyal, Rehman & Chauhan, 2010). Some studies report the presence of E. coli in salads (lettuce), vegetables and fruits and furthermore some strains have shown multidrug resistance capability (Solomon, Yaron & Matthews, 2002; Buchholz et al., 2011; Chekabab et al., 2013; Kim et al., 2014; Waturangi, Hudiono & Aliwarga, 2019). Recently, E. coli has been reported to be endophytic in plant leaves (Tharek et al., 2017). Some fragmentary reports are also available which show some plant growth promoting behaviour in E. coli (Nautiyal, Rehman & Chauhan, 2010; Sabat et al., 2014; Walker et al., 2013).

Plants grow in nature by making interactions with diversity of microbes including archaea, bacteria, fungi and algae. These microorganisms are abundantly present in soil and plants selectively recruit some of them as mutualistic beneficial partners. It is now accepted that plant growth promoting bacteria (PGPB) improve plant growth through nutrient acquisition and protection from pathogens (White et al., 2015, 2018, 2019). Phenotypic effects of soil rhizobacteria, rhizobia and mycorrhizae on plant growth and health have been well studied (Umali-Garcia et al., 1980; Zahran, 1999; Nautiyal, Rehman & Chauhan, 2010; Coba de la Peña et al., 2017). However, mechanisms of interaction, how they colonize on the surface or inside root tissues and, how they mobilize mineral nutrients from soil to plant roots are very limited. Observing tiny bacterial colonization on the surfaces and inside tissues of roots with their phenotypic effects on plant growth is difficult. We hypothesize that E. coli have capability to internalize into plant tissues naturally in the process of rhizophagy. Rhizophagy or the ‘rhizophagy cycle’ is a phenomenon by which plant roots cultivate bacteria on their root surface, internalize them into root cells and extract nutrients from them by subjecting them to superoxide produced by root cells (Paungfoo-Lonhienne et al., 2010, 2013; Soares et al., 2016; White et al., 2018). We further hypothesize that E. coli may set up a mutualistic relationship with plants and function endophytically to promote seedling development. In these symbiotic interactions microbes modulate plant development. The present study suggests the possibility of widespread occurrence of E. coli in vegetable crop plant tissues as a natural endophyte, rather than as a contaminant.

Materials and Methods

Bacterial strain

The bacterium used in this study is Escherichia coli -DH5-alpha (pPNptGreen) that expresses gfp gene with the nptII promoter which is resistant to 50 µg ml−1 of kanamycin, obtained from Dr. Gwyn Beattie at Iowa State University.

Plant material

Tomato (Solanum lycopersicum L.) and Bermuda grass (Cynodon dactylon L.) seeds were used in this study. Both plants seeds were procured from Hancock Farm & Seed Company and stored at 4 °C in a refrigerator.

Surface sterilization and disinfection

For surface sterilization and disinfection tomato and Bermuda grass seeds were first treated with 4% NaOCl (Clorox, Oakland, CA, USA) for 15 min with constant agitation on a shaker and washed several times with sterile distilled water. Then seeds were treated with 90% alcohol for 1 min and washed thoroughly with sterile distilled water until the scent of chlorine could not be detected (Verma & White, 2018). Effectiveness of surface sterilization and disinfection was checked by spreading last wash of disinfected seeds of tomato and Bermuda grass onto nutrient agar medium.

IAA (auxin) production, phosphate solubilization and antifungal activities of E. coli

Quantitative measurement of auxin (indole acetic acid-IAA) production from the bacterium was carried out following the protocol described by Gordon & Weber (1951) using Salkowski reagent (1 ml of 0.5 M FeCl3 and 50 ml of distilled water mixed with 30 ml of H2SO4). For that, bacteria were grown in LB broth supplemented with tryptophan and without tryptophan for 4 days with constant shaking. Four-day-old broth cultures were centrifuged for 2 min and 1 ml of supernatant was transferred into a fresh tube, then mixed with 2-ml of recently prepared Salkowski reagent and incubated for 30 min. Absorbance was recorded at 540 nm and quantification of IAA was done by comparing a standard curve of auxin as prescribed in previous study (Verma & White, 2018).

For the phosphate solubilization test, overnight grown cultures of E. coli were streaked onto Pikovskaya agar medium (Pikovskaya, 1948), and after 3–5 days of incubation, the colonies were observed for phosphate solubilization activity.

For antifungal activity, E. coli was tested for antagonism against common soil borne fungal pathogens including Fusarium oxysporum, Curvularia sp. and Alternaria sp. The experiment was done on PDA plates and the percentage inhibition of the radial growth of the fungus was calculated using the following formula described by Whipps (1997) [% inhibition = (R1 − R2/R1)]. Where R1 is control radial distance grown by the fungus and R2 is the distance grown by the fungus opposite to E. coli streaked on the same plate.

Inoculation experiment in Petri dishes

For initial observation of colonization and effects of bacterial inoculation on growth and development of Bermuda grass seedlings, the experiment was conducted on agarose and MS-agarose (Murashige and Skoog salt base in 0.7% agarose, wt./vol.) plates. For that, a total of 100 (50-50) surface disinfected Bermuda grass seeds were plated onto five agarose and five MS-agarose plates, respectively (10 seeds on each plate). Each seed was then inoculated with 5-µl Escherichia coli -DH5-alpha overnight grown suspension (10−5–10−8 bacteria ml−1). For controls, 50 seeds on agarose and 50 seeds on MS-agarose plates were inoculated with sterile distilled water. After 7 days of incubation, data for seedling health was recorded (see Table S2).

Inoculation experiment in magenta boxes

An experiment was setup in Magenta boxes containing sterilized peat, perlite and sand (2:2:1 ratio). In this experiment we used Bermuda grass and tomato. For that, first approximately 250 surface sterilized Bermuda grass seeds were inoculated with 5 ml of bacterial suspension (10−5–10−8 bacteria ml−1) for 2 h in sterile Petri plates. Similarly approximately 60 surface sterilized seeds of tomato were inoculated with 5 ml of bacterial suspension (10−5–10−8 bacteria ml−1) for 2 h in a sterile Petri plate. For controls, equal numbers of surface disinfected seeds of both species were inoculated with 5 ml water. After 2 h of treatment seeds were planted in magenta boxes. Around 150–180 Bermuda grass seeds treated with the bacterium and an equal number of untreated control seeds were planted in magenta boxes in triplicate (50–60 seeds in each box). Similarly, a total of 60 tomato seeds treated with the bacterium and 60 untreated control seeds were planted in magenta boxes in triplicate (20 seeds in each box). Data were recorded after 15 days of seedling growth.

Photosynthetic pigment measurements of Bermuda grass seedlings

For photosynthetic pigment measurement, chlorophyll a, b and carotenoids were estimated as per the method described by Smith & Benitez (1955). For that, 100 mg of fresh leaves of 15-day-old control and treatment seedlings were ground in liquid nitrogen and photosynthetic pigments were extracted into 15 ml of 80% acetone in triplicate. Acetone extracts were filtered through 0.45 µ filters and the optical density was recorded at various wavelengths, including 663, 645 and 440.5 nm. Chlorophyll a, b and carotenoids were measured by the following formulas:

TotalChlorophyllamgg−1freshleaf=12.7(O.D.)663−2.69(O.D.)645×(v/w×1000)

TotalChlorophyllbmgg−1freshleaf=22.9(O.D.)645−4.68(O.D.)663×(v/w×1000)

Totalcarotenoidsamgg−1freshleaf=46.95(O.D.440.5−0.268×chlorophylla+b)

where v = final volume (ml) of extract in 80% acetone, w = fresh weight of leaf in grams.

Fluorescence and confocal microscopy

The major aim of this study was to visualize the surface and internal colonization of bacteria during seedling development, and also determine whether bacteria were degraded in plant tissues. We used GFP-tagged E. coli and tracked them with fluorescence and confocal microscopy. Roots of E. coli inoculated Bermuda grass seedlings grown on agarose, MS agarose plates, and Bermuda grass and tomato seedlings grown in potting mix in magenta boxes were observed over several days to visualise green fluorescence from bacteria under fluorescence and confocal microscopy.

To visualize bacteria in seedlings, root tissues were observed using fluorescence microscopy on a Zeiss Axioskop® using the filter system for excitation at 485/498 nm. For confocal microscopy, root tissues were examined using a Zeiss LSM 710® confocal microscope using lasers 488 nm (for GFP) and 633 nm (for chlorophylls).

Statistical analysis

Microsoft Excel was used to analyse data and to prepare the bar diagrams. SPSS-16 program was used for t-tests to compare significance differences (P ≤ 0.05) in means of root and shoot lengths and photosynthetic pigments between the controls and treatments. All the means are presented with standard errors.

Results

Surface disinfection and primary screening of PGPR features

To check effectiveness of surface disinfection, the last wash (100 µl) of disinfected seeds of tomato and Bermuda grass were plated onto nutrient agar medium. None of the plates showed emergence of bacteria after incubation for two to three days at laboratory ambient temperature, thus surface disinfection was considered effective. Before treating seedlings, we evaluated bacteria for IAA (auxin) production, phosphate solubilization and antagonism against fungal plant pathogens. The bacterium was found to produce auxin in tryptophan amended and non amended nutrient broth medium with 8.68 ± 0.43 and 5.94 ± 0.24 µg ml−1 respectively (Table S1). The bacterium was also shown to be antagonistic against Fusarium oxysporum, Curvularia sp. and Alternaria sp. with 13%, 29% and 13% inhibition, respectively; however, it did not show phosphate solubilization activity (Table S1).

Initially, we screened the plant growth promoting features of E. coli on disinfected Bermuda grass seeds on agarose plates and MS-agarose plates. We found that treatment of the bacterium improved seedling health in terms of percent germination, positive geotropism, root-shoot length and number and length of root hairs (Table S2). Increased root lengths, and long abundant root hairs were observed in bacteria-treated Bermuda grass seedlings in comparison to control untreated (Table S2; Figs. 1 and 2).

Figure 1 Effect of inoculation of E. coli on seedling development of Bermuda grass and tomato 15 days after inoculation in magenta boxes containing peat, perlite and sand.

Where C is control, without inoculation and T is treatment with inoculation. (A–D) Bermuda grass; (E & F) tomato. Bar = 1 cm.

Figure 2 (A–F) Effect of inoculation of E. coli on root hair formation in Bermuda grass and tomato seedlings.

C, control; in both (Bermuda grass and tomato) very poor development of root hairs. T, treatment; in both (Bermuda grass and tomato) long and abundant development of root hairs (arrows).

Plant growth promotion in magenta boxes

In a magenta box experiments, inoculated seedlings of tomato and Bermuda grass showed improved growth and development (Fig. 1). Root lengths of 15-day-old seedlings of uninoculated (control) and inoculated (treatment) seedlings of Bermuda grass were recorded as 0.27 ± 0.09 and 0.43 ± 0.19 cm, respectively; for tomato 1.28 ± 0.38 and 2.39 ± 0.94 cm respectively; shoot lengths Bermuda grass control 1.81 ± 0.25 and treatment 1.90 ± 0.27 cm respectively; tomato shoots were 6.53 ± 0.82 and 6.68 ± 0.97 cm, respectively (Figs. 2 and 3). Similarly for photosynthetic pigments of Bermuda grass, Chl a for control and treatment were 0.53 ± 0.02 and 0.71 ± 0.02 mg g−1 fresh leaf weight respectively, for Chl b 0.24 ± 0.1 and 0.39 ± 0.03 mg g−1 fresh leaf weight, respectively, and for carotenoids 7.91 ± 0.20 and 9.53 ± 0.25 mg g−1 fresh leaf weight, respectively (Fig. 3).

Figure 3 Effect of inoculation of E. coli on root-shoot length in Bermuda grass and tomato, and photosynthetic pigments in Bermuda grass seedlings.

Inoculation significantly enhances the root length in both Bermuda grass and tomato seedlings (P < 0.001 and P < 0.003, respectively). However, there is no effect on shoot length but there are significantly increased photosynthetic pigments including chl a, chl b and carotenoids in Bermuda grass (P < 0.000, P < 0.009, and P < 0.001, respectively). Asterisks (* and **) and lowercase letters (a and b) show significant differences among the means.

Microscopy

The seedling roots of Bermuda grass inoculated with E. coli in MS Petri plates and magenta boxes, and tomato seedling roots in magenta boxes were microscopically (fluorescence) observed between 5 to 25 days of their growth. We found that GFP-tagged E. coli were heavily located around the root parenchyma of Bermuda grass as well as that of tomato seedling roots without any damage to the root parenchyma and root hairs (Figs. 4 and 5). Green fluorescing bacteria could be seen in the intercellular spaces of root tissues of both Bermuda grass and tomato. In some places of the root parenchyma, we observed the bacterial cells to change their morphology as they were degraded on root surface (Figs. 4C and 4D). In confocal microscopy, green fluorescing bacterial patches could be seen on the root parenchyma surface, intercellular spaces on both Bermuda grass and tomato roots (Figs. 4 and 5). Further, we also observed green florescence globular structures in vascular tissues of Bermuda grass roots (Fig. 4F).

Figure 4 Microscopic observation of Bermuda grass root tissue inoculated with GFP tag E. coli.

(A–D) Fluorescence microscopic images in which green fluorescing E. coli can be seen in the intercellular spaces of root parenchyma cells (A & B, arrows); degraded mass of green florescence structure can be also seen (C and D, arrows) at some places on root surface. (E and F) Confocal microscope images, where bacteria may be seen to colonize onto the surface and vascular tissues of the root (arrows). (G) The colonization of bacteria at the branch root tip (arrows).

Figure 5 Microscopic observation of tomato root tissue inoculated with GFP tag E. coli.

(A–E) Fluorescence microscopic images in which (A) and (B) are controls showing no bacterial colonization onto the root surface, and green fluorescing E. coli can be seen in the intercellular spaces of root parenchyma cells (C–E, arrows); degraded mass of green florescence structures can be also seen (C and D, arrows) at some places on root surface. (F–H) Confocal microscope images, where bacteria may be seen to colonize onto the surface. (H) Intact bacteria are not seen but some green fluorescing bacterial remains are observed.

Discussion

In this study, we sought to determine the interaction of a generally animal-associated microbe, E. coli, with Bermuda grass and tomato seedlings. Recently, many strains of E. coli have been reported from salad vegetables and fruits with virulence activities (Waturangi, Hudiono & Aliwarga, 2019). Ingestion of contaminated salad vegetables and fruits was proposed to be the reason for recent E. coli outbreak (Waturangi, Hudiono & Aliwarga, 2019). It has been suggested that strains of E. coli may become endophytic in plants and show growth promoting features (Paungfoo-Lonhienne et al., 2010; Nautiyal, Rehman & Chauhan, 2010; Walker et al., 2013; Tharek et al., 2017 ). One study demonstrated plant colonization in some phylogenetic groups of E. coli (Méric et al., 2013). Tharek et al. (2017) found E. coli strain USML2 to be an endophyte of inner leaf tissue of oil palm trees. Additionally, they found presence of several genes in E. coli used in plant colonization, including cellulase, pectinase, and plant growth promoting genes, including phosphate and potassium solubilising, and phytohormone-IAA biosynthesis genes. These traits suggest that E. coli may have capacity to colonize plants and degrade plant cell walls using pectinases and cellulases to enter plant cells and manipulate plant cell growth with hormones. These capacities may explain how Arabidopsis and tomato plants internalized E. coli into root cells in the process of rhizophagy (Paungfoo-Lonhienne et al., 2010), basically the bacteria degrade cell walls to enter root cells.

In the present study, we disinfected seeds of Bermuda grass and tomato by treating them with Clorox® and alcohol. However, occurrences of non-cultivable bacteria cannot be ruled out. The strain of E. coli we used showed several growth promoting features, including production of indole acetic acid (IAA), but was negative for phosphate solubilization. It demonstrated pathogen antagonistic activity, in inhibiting several fungal pathogens (Table S1). When we inoculated this bacterium onto disinfected Bermuda grass seeds on MS medium and in sterile potting mix in magenta boxes, and onto tomato seeds in potting mix in magenta boxes, we observed that inoculation improved the growth and health of the seedlings in comparison to controls (Fig. 1). Effects of inoculation can be seen more on root development in terms of greater numbers of roots, and longer roots and root hairs, and positive geotropism (roots growing downward). Root lengths of both Bermuda grass and tomato were significantly higher in E. coli treated seedlings than in controls (P < 0.001 and P < 0.003, respectively) (Figs. 1 and 3). However, there were no differences found in shoot length between treatment and control in both Bermuda grass and tomato seedlings (Fig. 3). All photosynthetic pigments, i.e., chlorophyll a, b and carotenoids, in Bermuda grass were found to be significantly reduced in non-inoculated controls compared to E. coli-inoculated seedlings (P < 0.000, P < 0.009, and P < 0.001, respectively) (Fig. 3). Increased root development in terms of root hairs and root length may be attributed to production of auxin, nitric oxide, ethylene or other signal molecules produced by E. coli within or in association with root hairs (Compant, Clément & Sessitsch, 2010; Verma & White, 2018; White et al., 2018; Chang, Kingsley & White, 2021). Plant growth promoting activity of some environmental strains of E. coli have been reported and characterized (Nautiyal, Rehman & Chauhan, 2010; Walker et al., 2013; Tharek et al., 2017). Nautiyal, Rehman & Chauhan (2010) suggested ubiquitous presence of plant growth promoting E. coli in soil as native soil bacteria rather than being and indicator of coliform contamination. Some reports also suggest that strains of E. coli may be endophytic (Sabat et al., 2014; Tharek et al., 2017). One finding reports that endophytic strains of E. coli produce cytokinin, even significantly more than Rhizobium and Bacillus species (Sabat et al., 2014).

When we observed seedlings 30 days after inoculation, we found E. coli-treated seedlings remained healthy, while uninoculated seedlings (controls) were yellowing and drying out. This was more visible with Bermuda grass seedlings (Fig. S1). That might be the result of continuous mobilisation and mineralisation of nutrients by E. coli in the rhizophagy cycle in plant roots (White et al., 2018, 2019). In the rhizophagy cycle, microbes alternate between a free-living phase in the soil, where bacteria gather soil nutrients, and an endophytic phase within root cells, where root cells extract nutrients from bacteria using reactive oxygen (superoxide). Bacteria enter plant root cells at the root tip meristem, situating in the periplasmic space (between root cell plasma membrane and the root cell wall) of root cells. Once internalized, the root cells secrete superoxide onto the bacteria, this causes bacteria to lose cell walls, forming naked protoplasts. Through continual cyclosis the root cells circulate the bacterial protoplasts around the periphery of the root cell, with the plasma membranes of the bacteria and root cells in continual contact, thus facilitating nutrient exchange between microbe protoplasts and root cells. Experiments have shown that bacterial protoplasts within root cells are replicated through a ‘budding’ process resulting in cloning multiplication of the initial bacteria. The presence of bacteria in root epidermal cells stimulates root hair elongation—with more bacteria present, hairs grow longer, and absence of bacteria results in failure of hairs to elongate (Verma et al., 2018; Chang, Kingsley & White, 2021). The bacteria within root cells have been hypothesized to produce ethylene that stimulates root hair elongation (White et al., 2019; Chang, Kingsley & White, 2021). As the root hairs elongate the microbes in hairs are ejected back into the soil where they may acquire additional nutrients and may later return to root tip meristems to once again enter root cells (White et al., 2019; Chang, Kingsley & White, 2021).

Through confocal and fluorescence microscopy we were able to visualize E. coli as they colonized Bermuda root tip meristem areas (Fig. 4). We observed that 8–10 day-old seedlings showed abundant colonization of the bacterium over the root surface and in intercellular spaces of both Bermuda grass and tomato seedlings (Figs. 4 and 5). Bacterial cells colonized the surface of root without any negative effect to the plant root cells. We were unable to visualize GFP fluorescence within most root epidermal cells, however some light fluorescence was seen in tomato root hairs (Fig. 5). In contrast, Paungfoo-Lonhienne et al. (2010) clearly demonstrated intracellular fluorescence of GFP-labelled E. coli in root epidermal cells and root hairs. Our reduced fluorescence may be due to use of a strain where GFP-expression is not strong, or because of reduced expression in the intracellular protoplast stage where bacteria are replicating rapidly and may not be expressing GFP as consistently as in the walled stage, or due to exposure to root cell-produced superoxide that may degrade GFP produced by the intracellular protoplasts. We were able to visualize globular green fluorescing structures in the vascular cylinder of Bermuda grass seedling roots (Figs. 4E and 4F). From the figure, it seems likely that these globular masses are clumps of E. coli cells in vascular tissues. These may be fluorescing because they may no longer be subjected to superoxide. Presence of E. coli inside vascular tissue suggests that the bacterium is capable of movement throughout plant tissues.

Conclusions

The results of our study show that plants may set up symbiotic associations with E. coli, where the bacterium is internalized into tissues of seedlings and may function to improve seedling development and nutrient status, and protect plants from disease agents. These findings are in contrast to the general assumption that E. coli is predominantly an animal-associated symbiotic or pathogenic bacterium or merely a contaminant of crop plants. It is also possible that the strains of E. coli in animal intestinal tracts may originate from E. coli strains in plants tissues. How extensively E. coli may be founds as an endophyte in plant tissues will require additional study.

Supplemental Information

Supplemental Information 1 One month old seedlings of burmuda uninoculated (control) and inoculated with E. coli (treatment).

Click here for additional data file.

Supplemental Information 2 IAA (auxin) production by E. coli (GFP) in LB broth medium, Antifungal activity against fungal pathogen, Phosphate solubilisation activity and plant growth promotion by E. coli.

Supplementary Table 1. IAA (auxin) production by E. coli (GFP) in LB broth medium with and without tryptophan, antifungal activity against F. oxysporum, Curvularia sp. and Alternaria sp., and phosphate solubilisation.

Supplementary Table 2. Initial screening of plant growth promoting features of E. coli (GFP) on Bermuda onto agarose and MS- agarose media.

Click here for additional data file.

Supplemental Information 3 Seedling root data: shoot growth and photosynthetic pigments.

1. Raw data: Effect of inoculation of E. coli (treatment) on root-shoot length in tomato seedlings.

2. Raw data 2: Effect of inoculation of E. coli (treatment) on root-shoot length in Bermuda seedlings.

3. Raw data: For photosynthetic Pigments (Chl a, b and carotenoids) (mg g−1 fresh weight).

Click here for additional data file.

Additional Information and Declarations

Competing Interests

Author Contributions

Data Availability

The authors declare that they have no competing interests.

Satish K. Verma conceived and designed the experiments, performed the experiments, analyzed the data, prepared figures and/or tables, authored or reviewed drafts of the article, experimentation, Microscopy, and approved the final draft.

Qiang Chen performed the experiments, prepared figures and/or tables, microscopy, and approved the final draft.

James Francis White conceived and designed the experiments, performed the experiments, analyzed the data, prepared figures and/or tables, authored or reviewed drafts of the article, and approved the final draft.

The following information was supplied regarding data availability:

Raw data are available in the Supplemental Files.

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
