# Peer review of "Evaluation of colonization and mutualistic endophytic symbiosis of Escherichia coli with tomato and Bermuda grass seedlings"

_PeerJ, doi:10.7717/peerj.13879_

## Round 0.1 · original submission · Major Revisions

Dear Dr. Verma and colleagues:

Thanks for submitting your manuscript to PeerJ. I have now received four independent reviews of your work, and as you will see, the reviewers raised some concerns about the research. One reviewer raised serious concerns about the research and recommended rejection; however, the other three reviewers are more enthusiastic and raised some areas that need substantial improvement. I agree with the concerns of all reviewers.

I am affording you the option of revising your manuscript according to all four reviews but understand that your resubmission may be sent to at least one new reviewer for a fresh assessment (unless the reviewer recommending rejection is willing to re-review).

Importantly, please address the concerns raised by reviewer 4 regarding the lack of statistics. I find this concerning and will expect and strong response to this criticism.

The majority of these criticisms is highly constructive and should immensely improve your manuscript. Please also note that reviewers 3 and 4 have included marked-up versions of your manuscript.

Please address all of these concerns in your rebuttal letter. Also, please have an English expert help proof your revision.

Good luck with your revision,

-joe

·

Basic reporting

Present study titled Evaluation of colonization and mutualistic endophytic symbiosis of Escherichia coli with tomato and Bermuda grass seedlings- describe about the successful colonisation of E. coli in tomato and Bermuda roots. Bacterium has also shown some plant growth promoting activity also. With confocal microscopy author have tried to confirm its endophytic colonisation in roots of tomato and Bermuda. Study is interesting since E. coli is generally animal gut bacterium and reported as water bodies’ contaminant. Paper is written well some of the comments and questions are given below which need to be addressed by the authors.

Experimental design

1. In the material method section authors have not given details about the microscopy procedure-
This section needs revision. The introduction part may also be elaborated.
2. Why have the authors measured only photosynthetic pigments from Bermuda leaf not from tomato, for any specific reason?
3. Authors have shown that E. coli inoculation induces the root growth in tested plants but no effect on shoot- why. It must be discussed appropriately.
4. Did authors examined the shoots or leaves of the plant for the endophytic colonisation of bacterium? It would very interesting to add it.
5. Authors may check for typose error at some places in the manuscript.

Validity of the findings

Line 137-138- This should be in the materials and method section.
Line 143-144- The bacterium is producing more auxin in tryptophan less medium when compared to tryptophan amended medium. Can there be a reason as it’s an interesting result?
Line 194-196- The authors mention that presence of cellulase and pectinase activities might help in the successful colonization of E. coli inside the host. However, the authors did not check whether these activities are present in their strain or not. Since the authors have shown successful colonization as well as rhizophagy, therefore it would have been interesting to see
whether they possess these traits or not?
Line 197- The authors have used the term disinfected, but in my opinion, it should be surface sterilized as the presence of endophytic bacteria cannot be ruled out by chlorox treatment. Reports have shown in vitro activation of endophytic bacteria after surface sterilization
(Refer to Shaik and Thomas, 2019 “In Vitro Activation of Seed-Transmitted Cultivation-Recalcitrant Endophytic Bacteria in Tomato and Host–Endophyte Mutualism”).
Line 205-206- The growth effects of the strain are seen mainly on roots and not on shoots. However, the authors might have checked the dry weight of shoots or the whole seedlings as better root architecture generally leads to better plant growth.
Line 226-228- The authors attribute better mineralization and mobilization of nutrients contributing to healthy E. coli treated seedlings but the strain showed no phosphate solubilization activity while potassium solubilization, nitrogen fixation, siderophore production was not checked. The authors did mention the rhizophagy cycle; however, a detailed study of the cycle is required to confirm its role in nutrient mineralization and solubilization.

Additional comments

Authors may check for typose error at some places in the manuscript.

Reviewer 2 ·

Basic reporting

The study is interesting but some revisions are needed
Line 51 rhizophagy
239 reference is wrongly written
272 acknowledge
Some references are missing
Figure 3 looks old school style
Figure 4 is horrible, please delete it or do it again by reducing aperture or by decreasing the time of talking the photo
Figures 4-8 can be combined in a single figure, and it will be better for the reader.
Figure 8 confocal microscopic observations, where

Experimental design

Do the authors always used gfp trasnformed cells? Normally we compare to wild type strain for effects on plants, but is also ok.

Validity of the findings

The study is interesting but some revisions are needed
Line 51 rhizophagy
239 reference is wrongly written
272 acknowledge
Some references are missing
Figure 3 looks old school style
Figure 4 is horrible, please delete it or do it again by reducing aperture or by decreasing the time of talking the photo
Figures 4-8 can be combined in a single figure, and it will be better for the reader.
Figure 8 confocal microscopic observations, where

Reviewer 3 ·

Basic reporting

The text is clear but can be improved.
The article is conforming to professional expression standards.
The work is simple in methodology, reaching an important contribution by demonstrating that Escherichia coli can establish as an endophyte in at least two types of plants, becoming a plant-symbiont by stimulating the host development.
Literature references are sufficient in this field background and well contextualized.
The article includes sufficient introduction and background to demonstrate how the work fits into the broader field of knowledge. Relevant prior literature is appropriately referenced. Nevertheless, text needs edition from a fluent English-speaking microbiologist or microbial ecologist.

Experimental design

The adopted methodology is correct and well described, with the necessary and sufficient details to be repeated. The experiments were well planned and met the assumptions of casual accuracy. Statistical analysis is missing in M&M section of results. Nonetheless, statistical values are described in the discussion section (line 208)

Validity of the findings

The results are conclusive and supported by the appropriate methodology, including statistical analysis. The hypotheses have been tested and proven.
Conclusions are well stated, linked to original research question & limited to supporting results.
Nevertheless, authors failed describing the statistical analysis used method.

Additional comments

The article is valuable, demonstrating that Escherichia coli can establish as an endophyte in at least two types of plants, and become a plant-symbiont by stimulating the host development. Thus, this study has great importance and applicability.
Nevertheless, text needs edition from a native English microbiologist or microbial ecologist. It is recommending to refer in the text, first the older reference and the newer references at the end, taking in consideration also the alphabetical order. Per example in line 40, change “(Walker et al., 2013; Nautiyal et al., 2010; Sabat et al., 2013)” by “(Nautiyal et al., 2010; Sabat et al., 2013; Walker et al., 2013).
Throughout the text, many “commas” are missing.

Annotated reviews are not available for download in order to protect the identity of reviewers who chose to remain anonymous.

Reviewer 4 ·

Basic reporting

1. Please homogenize your English. Some places e.g., colonise (L47), mobilise (L48), colonisation (L49), etc. use British English, and other places, e.g., colonize (L21) and internalize (L51), use American English.

2. L12. Please spell “E. coli” out.

3. L15. Please show the scientific name of tomato and Bermuda grass here.

4. L95. A total of.

Experimental design

1. Please provide the objectives of this study in the introduction.

2. The authors concluded that effects on seedling root growth could in part be explained by IAA production. IAA indeed is able to promote plant growth, but this conclusion drawn based on a culture-based study is speculative.

Validity of the findings

1. This study has a significant limitation: no statistical analyses of data were present. Without statistical analyses, the conclusions you reached are not convincing.

2. L69-72. Please add the refs for the surface sterilization you used.

Annotated reviews are not available for download in order to protect the identity of reviewers who chose to remain anonymous.

---

## Round 0.2 · Minor Revisions

Dear Dr. Verma and colleagues:

Thanks for revising your manuscript. The reviewers are very satisfied with your revision (as am I). Great! However, there are a few additional concerns to address. Please attend to these issues ASAP so we may move towards acceptance of your work.

Best,

-joe

Reviewer 2 ·

Basic reporting

Authors improved their manuscript as suggested. It is ready for publication. just a minor mistake: Line 72 seeds instead of seed

Experimental design

Authors improved their manuscript as suggested. It is ready for publication. just a minor mistake: Line 72 seeds instead of seed

Validity of the findings

Authors improved their manuscript as suggested. It is ready for publication. just a minor mistake: Line 72 seeds instead of seed

Reviewer 3 ·

Basic reporting

This is the reviewed version. Authors addressed each question and comment correctly.
Manuscript is approved.

Experimental design

In the M&M section, significant value for differences is missing. I suggest to insert “(P ≤ 0.05)” after “significance differences” (line 147)

Validity of the findings

The results are conclusive and supported by the appropriate methodology, including statistical analysis. The hypotheses have been tested and proven.
Conclusions are well stated, linked to original research question & limited to supporting results.

In results section, change “P≤ 0.001 and 0.003 respectively” by “P < 0.001 and P < 0.003, respectively” (line 220); and “(P≤0.000, 0.009, 0.001 respectively).” by “(P < 0.001, P < 0.009, and P < 0.001, respectively).” (line 224)

In Figure 3 legend, Change “P≤ 0.001 and 0.003 respectively” by “P < 0.001 and P < 0.003, respectively”; and “(P≤0.000, 0.009, 0.001 respectively).” By “(P < 0.001, P < 0.009, and P < 0.001, respectively).”

Additional comments

Be aware that the worlds in manuscript and book chapter titles, the first letter should not be written in capital letter (such as in Tharek et al. and Walker et al. references). That is only for book title.

Reviewer 4 ·

Basic reporting

The manuscript is improved a lot compared to the previous one.

Experimental design

Looks good.

Validity of the findings

Looks good.

Additional comments

Fig. 3: Please show the significant differences between controls and treatments directly using letters or asterisks.

---

## Round 0.3 · accepted · Accept

Dear Dr. Verma and colleagues:

Thanks for revising your manuscript based on the concerns raised by the reviewers. I now believe that your manuscript is suitable for publication. Congratulations! I look forward to seeing this work in print, and I anticipate it being an important resource for groups studying plant endophytes. Thanks again for choosing PeerJ to publish such important work.

Best,

-joe